# A conserved ankyrin repeat-containing protein regulates conoid stability, motility and cell invasion in *Toxoplasma gondii*

Shaojun Long[1], Bryan Anthony[1], Lisa L. Drewry[1] & L. David Sibley [1]

Apicomplexan parasites are typified by an apical complex that contains a unique microtubule-organizing center (MTOC) that organizes the cytoskeleton. In apicomplexan parasites such as *Toxoplasma gondii*, the apical complex includes a spiral cap of tubulin-rich fibers called the conoid. Although described ultrastructurally, the composition and functions of the conoid are largely unknown. Here, we localize 11 previously undescribed apical proteins in *T. gondii* and identify an essential component named conoid protein hub 1 (CPH1), which is conserved in apicomplexan parasites. CPH1 contains ankyrin repeats that are required for structural integrity of the conoid, parasite motility, and host cell invasion. Proximity labeling and protein interaction network analysis reveal that CPH1 functions as a hub linking key motor and structural proteins that contain intrinsically disordered regions and coiled coil domains. Our findings highlight the importance of essential protein hubs in controlling biological networks of MTOCs in early-branching protozoan parasites.

[1] Department of Molecular Microbiology, Washington University School of Medicine, St. Louis, MO 63110-1093, USA. Correspondence and requests for materials should be addressed to L.D.S. (email: sibley@wustl.edu)

The phylum Apicomplexa is a diverse group of protozoans that includes a large number of both free living and parasitic species[1]. Apicomplexans cause important human diseases, such as malaria caused by *Plasmodium* spp., toxoplasmosis caused by *Toxoplasma gondii* and cryptosporidiosis by *Cryptosporidium* spp. The phylum is unified by a unique microtubule-organizing center (MTOC) that consists of a polar ring complex that collects singlet microtubules that subtend the membrane[2]. The apical complex coordinates protein secretion, gliding motility, active invasion of, and egress from host cells[3]. The apical complex has no analogous structure in higher organisms, and it is largely known from morphological studies, which originally formed the basis for the classification of organisms into this phylum[4]. The conoid and the cortical cytoskeleton also play key roles in the unique modes of cell division by apicomplexans[5,6].

In a subset of apicomplexans, known as coccidians, the apical complex is further specialized by the presence of a microtubule-rich structure called the conoid[2]. In *T. gondii*, a model coccidian parasite, the apical complex consists of two pre-conoidal rings, the conoid that forms a cone of spirally arranged tubulin-rich fibers, and an apical polar ring at the base that anchors the 22 singlet, subcortical microtubules[7,8]. The spiral conoid can extend beyond the apical polar ring, or retract below it, a process controlled by intracellular calcium and the actin cytoskeleton[9]. The apical polar ring lies at the boundary of the inner membrane complex (IMC), a flattened system of membranes that lies beneath the plasma membrane and which consists of a system of interconnecting plates that extend backward, culminating in the basal complex[10]. Motor proteins that are crucial for motility exist within the conoid (e.g., MyoH)[11] or anchored in the IMC and extending into the space between the IMC and the plasma membrane (e.g., MyoA)[12].

The apical polar ring is highly conserved, while the complexity of the conoid varies among the apicomplexans[2]. For example, a highly developed conoid is found in coccidians, such as *Eimeria* spp., and *Neospora* spp. *Sarcocystis* spp., which share a higher proportion of orthologs, relative to other members of the phylum[13]. Additionally, *Cryptosporidium* spp., a deep branching apicomplexan, contains a fully formed conoid[14], suggesting this is a shared ancestral trait. However, other apicomplexan species such as *Plasmodium* spp. lack a proper conoid but retain the apical polar ring[2] and components of the pre-conoidal rings, such as SAS6L[15], suggesting the apical complex is condensed in the absence of the tubulin-rich conoid. Relatives of apicomplexans (e.g., *Colpodella* and *Perkinsis*)[16] and *Chromera velia*[17] contain a half-closed cone structure (pseudo-conoid) at their apical end, which may be related to the conoid seen in some apicomplexans. Among these diverse species, *T. gondii* provides a model for dissecting the structure and function of the conoid.

The conoid in *T. gondii* is comprised of 14 tubulin-rich fibers that do not form conventional closed microtubules, but rather adopt a comma shape[7]. This asymmetric structure consists of nine tubulin protofilaments[7] and recent studies implicate a microtubule-binding protein called DCX in controlling this unusual tubulin polymer[18]. DCX contains double cortin and P25-α domains, which bind to microtubules in other organisms to regulate stability. Deletion of DCX results in a shorter conoid and defects in invasion[18]. Other apical proteins found in the apical polar ring include RNG1, which is refractory to genetic disruption[19], and RNG2, loss of which affects microneme secretion[20]. Additionally, double deletion of kinesin A and apical polar ring protein 1 (APR1) results in fragmentation of the apical polar ring and a defect in microneme secretion[21]. A previous proteomic study revealed that >250 proteins were enriched in the conoid of *T. gondii*[8], suggesting a much greater complexity of this structure than is currently appreciate by studies of individual components.

Here we compare the predicted conoid proteome to the transcriptional profile of known apically located proteins, and use epitope tagging to verify 11 of 21 candidates as authentic apical complex proteins. CRISPR/Cas9-mediated deletion of genes encoding these apical proteins identifies a single essential ankyrin repeat-containing protein that is conserved in the Apicomplexa. This ankyrin repeat-containing protein is essential in *Toxoplasma* for invasion, egress, and motility, despite not affecting microneme secretion. Proximity labeling using permissive biotin ligase fusions reveals that this protein acts as a conoidal protein hub (CPH1) at the center of a protein interactome that is important for controlling the structural integrity of the conoid.

## Results

### Discovery of apically localized proteins in *T. gondii*. To identify new components of the apical complex in *T. gondii*, we compared the transcriptional profiles for known apical complex proteins, based on a previous analysis of cell cycle expression[22]. Comparison of these candidates revealed similar profiles with peak levels during S to M phase that dropped during cytokinesis and reach their lowest level in G1 phase (Fig. 1a). We then manually compared this expression profile to the predicted proteome data for the conoid defined previously[8]. We identified 30 hypothetical proteins that fit these criteria and removed those found in other compartments, thus narrowing the list to 21 proteins (Fig. 1a, b) (Supplementary Table 1). To localize these candidates, we used CRISPR/Cas9 to add 6HA epitope tags to the C termini (Supplementary Tables 1–4), as described previously[23,24]. Immunofluorescence staining confirmed that 11 of tagged proteins were localized to the apical end, while one was partially apical, and another 10 localized to other compartments (Supplementary Fig. 1 and Supplementary Tables 1, 5). We next tested the essentiality of the 11 apically localized proteins using a double CRISPR single guide RNA strategy to delete the genes, as previously described[24,25]. We were easily able to delete 10 of the 11 genes, as confirmed by diagnostic PCR (Supplementary Fig. 2a), and these knockouts had no discernable (9) or mild (1) growth defects in vitro (Supplementary Fig. 2b and Supplementary Tables 1–5). One gene (TGME49_266630, Supplementary Tables 1–5) was refractory to CRISPR/Cas9 deletion, suggesting it was essential. Based on features described below, this protein was named Conoid Protein Hub 1 (CPH1).

### Ankyrin repeat-containing protein CPH1 is localized to the conoid. Although the initial IFA results indicated that CPH1 localized to the apical end of the parasite, we could not be certain of its precise location based on conventional light microscopy (Supplementary Fig. 1a). Therefore, we utilized super-resolution microscopy to examine the distribution of CPH1 in cells where the conoid was protruded and compared its pattern to known proteins, RNG2[20] and CaM1[23]. To facilitate precise colocalization, we generated lines where the C-terminus of CaM1 and N-terminus of RNG2 were tagged with the Ty epitope (2Ty) in the CPH1-6HA line using CRISPR/Cas9 (Supplementary Tables 2–4). Super-resolution microscopy revealed that CPH1-6HA was concentrated in a narrow band within the protruded conoid (Fig. 1c). Staining for CPH1-6HA labeled a broad region of the conoid that partially overlapped with both reference proteins (Fig. 1c). Consistent with this pattern, immunoEM revealed that CPH1-6HA was concentrated in the conoid, slightly ahead of the position of the extended conformation of RNG2 (Fig. 1d). CPH1 contains two ankyrin domains in *T. gondii*, and this feature is conserved in

apicomplexans that have a complete conoid, with the exception of *Eimeria* and *Cryptosporidium*, where the ankyrin repeats are degenerate (Fig. 1e). Other members of the phylum, including the early-branching gregarines, also contain orthologs of CPH1, although they do not contain the conserved ankyrin repeats (Fig. 1e).

**CPH1 is essential for parasite motility, invasion, and egress.** To dissect the function of CPH1, we constructed a conditional knockdown CPH1-AID line using the auxin-inducible degron system, previously developed for studying essential genes in *T. gondii*[23,26]. CPH1-AID was expressed in the TIR1 line, which is required for controlled degradation, as confirmed by western

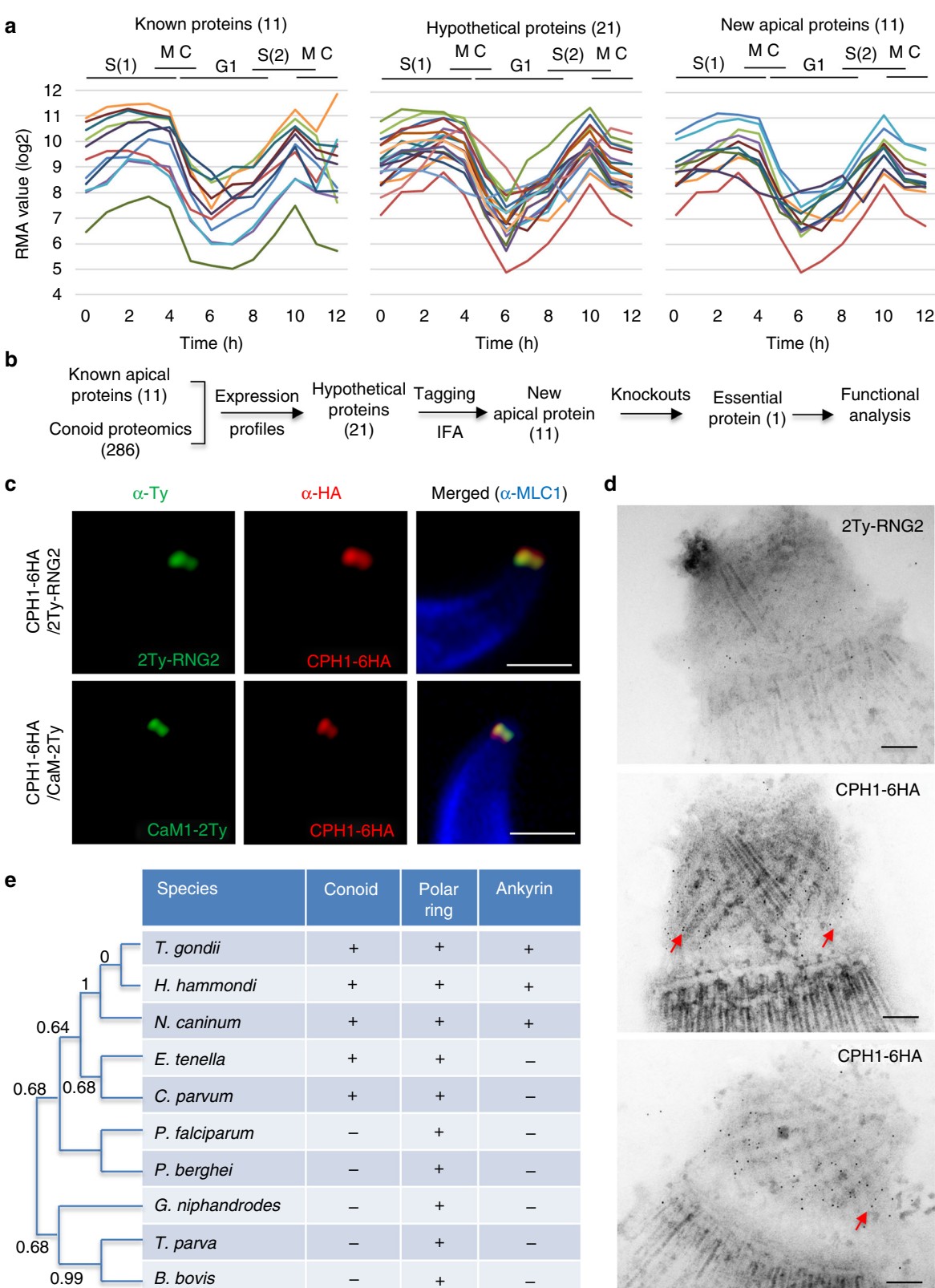

blot (Fig. 2a). Addition of auxin resulted in degradation of CPH1-AID to <5% within 4 h, as shown by western blot (Fig. 2b, Supplementary Fig. 3a). CPH1-AID was correctly localized to the apical end, and addition of auxin resulted in its degradation, as shown by IFA (Fig. 2c). To test the essential nature of CPH1, we examined the ability of parasites to form plaques on host cell monolayers. Both the TIR1 and CPH1-AID lines grew normally in absence of auxin, whereas addition of auxin led to a complete block in plaque formation in the CPH1-AID line (Fig. 2d). However, treatment of the CPH1-AID line with auxin did not affect replication during 24 h (Fig. 2c, Supplementary Fig. 3b), although there was an increase in the number of vacuoles containing ≥8 parasites, which may result from an egress defect, as described below. Treatment with auxin was effective over a range of concentrations from 25–2000 μM without adverse effects on growth of the parental TIR1 or the host cell monolayer (Supplementary Fig. 3c–f).

Plaque formation requires parasite motility, invasion, and egress, steps that depend on adhesins secreted from micronemes[27]. Therefore, we tested whether microneme secretion was affected upon depletion of CPH1. We used the secretory microneme protein 2 fused to *Gaussia* luciferase (MIC2-GLuc) as a reporter for agonist-induced secretion, as described previously[28]. We observed that secretion of MIC2-GLuc by extracellular parasites was normal following depletion of CPH1 by growth in auxin (Fig. 2e). However, video microscopy showed that extracellular parasites were significantly impaired in motility under similar conditions (Fig. 2f). Consistent with impaired motility, parasite invasion was strongly inhibited (Fig. 2g). In intracellular parasites, microneme secretion is also required for parasite egress from host cell, and the initial steps in this process are mediated by a release of TgPLP1, which disrupts the parasitophorous vacuole[29]. To monitor microneme secretion by intracellular parasites, we tested the permeabilization of vacuole by diffusion of a secreted form of DsRed, as described previously[27]. DsRed rapidly diffused from the parasitophorous vacuole into cytoplasm of host cells upon treatment of the CPH1-AID line with calcium ionophore, both in the presence and absence of auxin (Fig. 2h, i). These findings demonstrate that microneme secretion occurs normally in intracellular parasites following depletion of CPH1. However, ionophore-induced parasite egress from host cells was completely blocked under these conditions (Fig. 2j).

**CPH1 is required for conoid stability in extracellular parasites**. As CPH1 was exclusively localized to the conoid, we wondered if there were any defects in formation of the conoid following depletion of the protein. We examined the conoid in intracellular replicating parasites by transmission electron microscopy (TEM) in CPH1-AID parasites that were grown in the absence or presence of auxin. There were no obvious changes in the tubulin-rich, protein fibers that form the spiral shaped conoid (Fig. 3a),

indicating the conoid assembly was normal. We then assessed if the conoid can protrude in extracellular parasites following stimulation with calcium ionophore, which is known to activate this process[9]. CPH1 depletion had no effect on conoid protrusion (Fig. 3b). Control and auxin-treated CPH1-AID parasites were detergent extracted and examined by negative staining and transmission EM, a process that reveals the conoid architecture[7,8]. In parasites where CPH1 was degraded, the conoid was shortened and partially collapsed (Fig. 3c). Quantification of these effects (as shown in Fig. 3c) showed that upon depletion of CPH1, the conoid was reduced in length (d), and in width at the anterior (a) (the pre-conoid rings), middle (b), and posterior regions (c) (the apical polar ring) (Fig. 3d).

**Ankyrin repeats are critical for targeting and function of CPH1**. CPH1 is predicted to have two ankyrin repeats, a domain that is often involved in protein–protein interactions. Ankyrin domain 1 is more conserved and extends from residues 31 to 114 (core, 48–91), while ankyrin domain 2 is more degenerate and extends from residues 429 to 463 (Fig. 4a). To assess the functional role of ankyrin repeats in CPH1, we introduced deletions to the core ankyrin 1 (CPH1$^{\Delta ank1}$-Ty) and ankyrin 2 (CPH1$^{\Delta ank2}$-Ty) domains and generated complementing lines in the CPH1-AID background with these mutants and a wild-type CPH1-Ty construct (Fig. 4a). Complementation with a full-length CPH1-Ty construct resulted in targeting to the conoid, both in the presence and in the absence of auxin, and this construct fully restored growth when CPH1-AID was degraded (+indoleacetic acid (IAA)) (Fig. 4b). In contrast, CPH1$^{\Delta ank1}$-Ty was mistargeted to the cytosol in the absence of auxin, but was partially re-localized to the apical end, and was able to complement the CPH1-AID line grown in the presence of auxin (Fig. 4b). In contrast, transiently expressed CPH1$^{\Delta ank2}$-Ty remained in the cytosol when the endogenous copy was depleted with auxin treatment (Fig. 4c). We were not able to generate a stable line of CPH1$^{\Delta ank2}$-Ty, suggesting this construct has a dominant negative effect. Consistent with these findings, TEM examination of negatively stained cytoskeletons revealed that the structural defects in the conoid were fully rescued by complementation with wild-type CPH1 and partially with CPH1$^{\Delta ank1}$-Ty, in the presence of auxin (Fig. 4d). These findings suggest that CPH1 is targeted to the conoid by interacting with proteins through the ankyrin repeat regions, and that these interactions are essential for conoid stability in extracellular parasites. Alternatively, it is possible that the ankyrin repeat regions are needed for proper folding and that other domains mediate the interaction of CPH1 with the conoid.

**CPH1 is a central hub for interaction in the conoid**. To identify proteins that interact with CPH1 in the parasite, we used a permissive biotinylation approach that detects proteins that are proximal to a bait protein fused with the biotin ligase BirA[30]. This approach was chosen over conventional immuno-precipitation as

**Fig. 1** Discovery of new proteins localized to the apical complex in *T. gondii*. **a** Comparisons for known proteins previously localized to the apical complex (left), new hypothetical proteins predicted (middle), and proteins identified to be at the apical complex (right). Patterns based on expression pattern during the cell cycle[49]. Known proteins included: RNG1[19], RNG2[20], AKMT[50], MyoH[11], CaM1, CaM2, CaM3[23], MLC3, MLC5, MLC7[11], SAS6L[39]. **b** Workflow used for identification, confirmation, and functional analysis of new apical proteins with protein numbers shown at each step. **a**, **b** See also Supplementary Table 1 and Supplementary Figs. 1, 2. **c** Super-resolution imaging of the essential protein CPH1 at the conoid. IFA was performed with mouse anti-Ty (green), rat anti-HA (red), and rabbit anti-MLC1 (blue) followed by secondary antibodies conjugated to Alexa Fluor dyes. Scale bar = 1 μm. **d** Localization of CPH1 at the apical complex by immunoelectron microscopy. Red arrows indicate distribution of gold particles along the microtubules. Scale bar = 100 nm. **c**, **d** Extracellular parasites were stimulated with 3 μm A23187 for 10 min to extend the conoid prior to processing. Images are representative of three or more experiments with similar outcomes. **e** Conservation of CPH1 and the apical complex in apicomplexan parasites. CPH1 orthologs (Supplementary Table 1 lists genes and species names) were used to generate the dendrogram shown (left). The chart summarizes the presence (+) and absence (−) of the conoid or polar ring[2] and ankyrin repeats in CPH1 (present study)

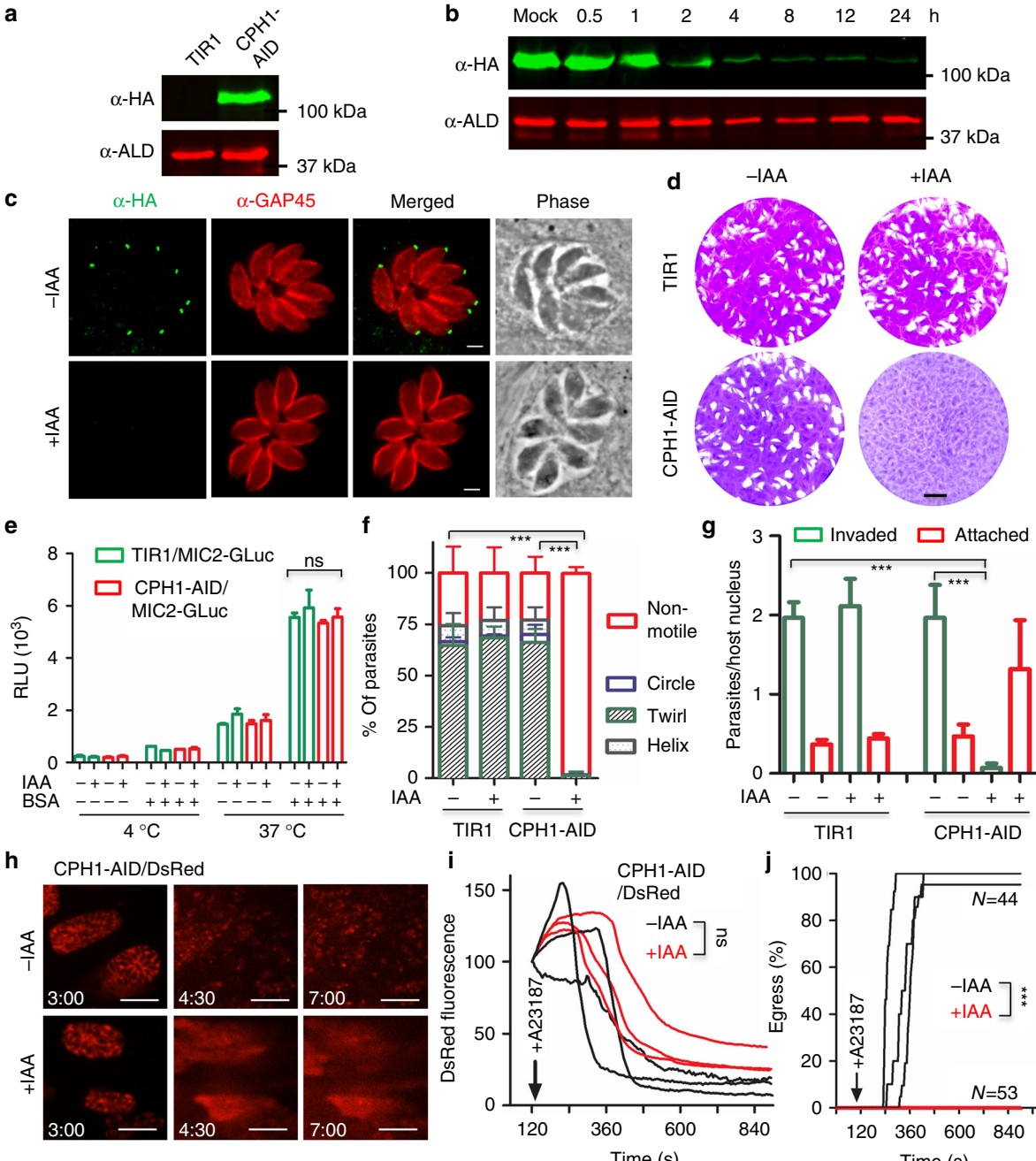

**Fig. 2** CPH1 is essential for parasite motility, invasion, and egress but not for microneme secretion. **a** Western blot analysis of CPH1 endogenously tagged with AID-3HA (α-HA) in the parental TIR1 line (shown for comparison). Aldolase (α-ALD) loading control. **b** Degradation of CPH1-AID triggered by addition of 500 μM auxin (+IAA) for different times (h), vs. mock, 0.1% ethanol (−IAA). Western blot performed as in **a** (Supplementary Fig. 3A). **c** Immunofluorescence microscopy of CPH1-AID parasites grown ± IAA for 24 h stained using mouse anti-HA (green) and rabbit anti-GAP45 (red). Scale bar = 2 μm. **d** Plaque formation with TIR1 and CPH1-AID lines grown ± IAA for 7 days. Scale bar = 0.5 cm. **a**–**d** Representative of three or more experiments with similar outcomes. **e** Microneme secretion following stimulation with bovine serum albumin (BSA) plus 1% ethanol for 10 min. **f** Parasite motility was recorded by video microscopy, manually scored, and plotted as percentage of total. ***$P < 0.0001$. **g** Parasite invasion into HFF cells was examined after 20 min challenge. ***$P < 0.0001$. **e**–**g** Parasite lines were grown in ±IAA for 40–44 h and extracellular parasites were collected and immediately used for assays. Mean ± SEM ($n = 3$ experiments, each with 3 technical replicates, $n = 9$). One-way ANOVA with Kruskal–Wallis test (**e**) or with Tukey's multiple comparison (**g**) or two-way ANOVA (**f**) with Tukey's multiple comparison. **h**–**j** Parasite lines were grown in ±IAA for 36 h and video microscopy was performed after addition of 2 μm A23187. **h** Representative example of images from three independent experiments showing time lapse recordings of DsRED diffusion (time stamp min:s). Scale bar = 20 μm. **i** Mean of replicates ($n = 8$ vacuoles) from each experiment plotted as separate lines ($n = 3$ experiments). **j** Mean of replicates ($n = 44$ −IAA, 53 +IAA in total) from each experiment plotted as separate lines ($n = 3$ experiments). ***$P < 0.0001$. Two-way ANOVA with Tukey's multiple comparison. RLU relative luminescence units, ns not significant

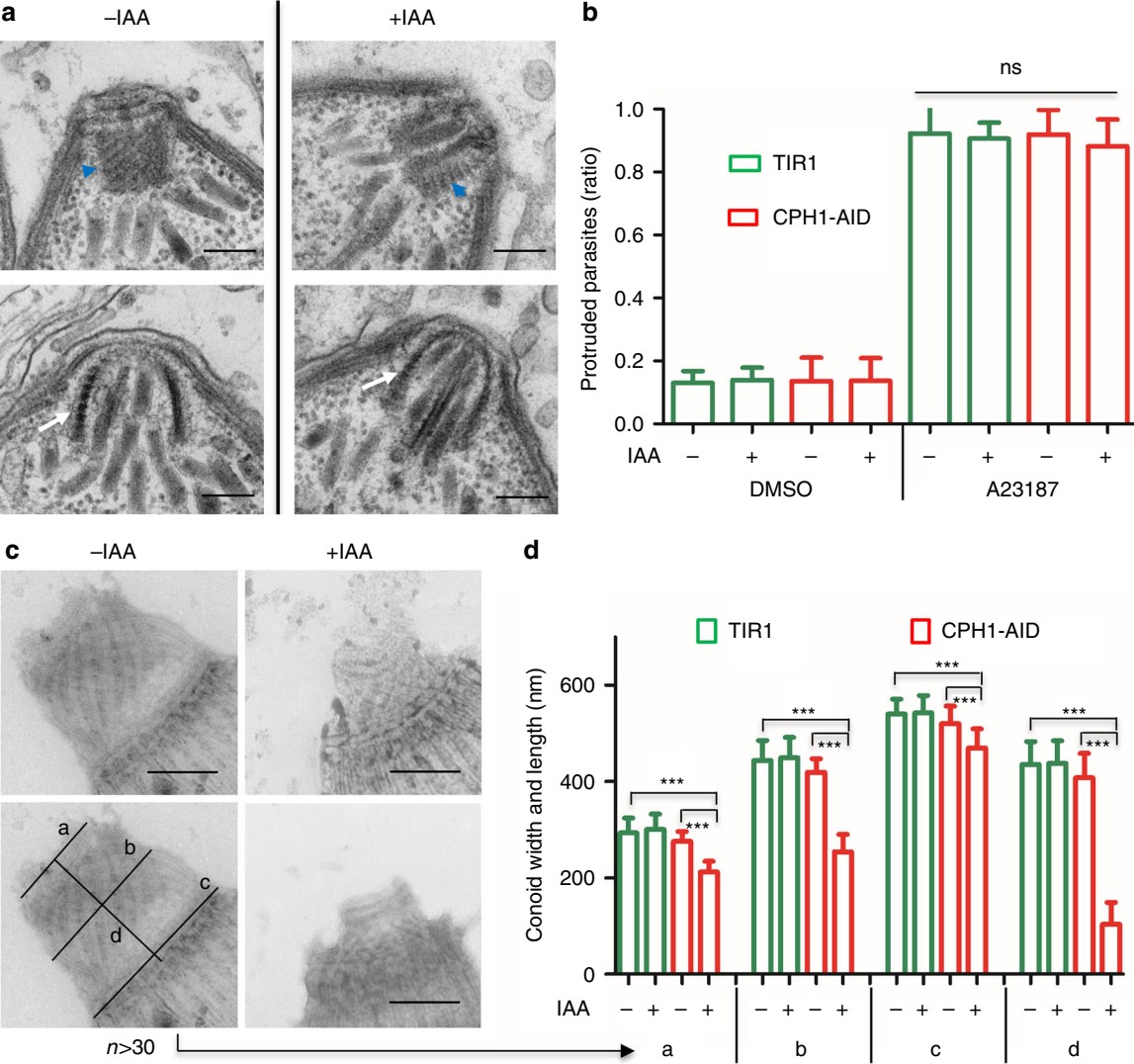

**Fig. 3** CPH1 is required for conoid stability in extracellular parasites. **a** Conoid morphology of intracellular parasites examined by TEM. CPH1-AID parasites grown in ±IAA for 24 h. Microtubule spirals seen in tangential sections (blue arrowhead) or cross-section (white arrow). Scale bar = 200 nm. **b** Conoid protrusion in extracellular parasites as assessed under phase contrast microscopy. Mean ± SEM for 3 experiments with 3 replicates for each ($n = 9$). DMSO served as a control. *ns* not significant, one-way ANOVA. **c** Conoid morphology of extracellular parasites examined by negative staining and TEM. Image in left bottom shows measurements for the conoid width and length (labeled as **a–d**). The line with arrow points to the measurements in **d**. Scale bar = 250 nm. **d** Measurement of conoid width and length as shown in **c**. ***$P < 0.0001$, mean ± SEM ($n = 3$ experiments, each with 3 technical replicates), $n = 35$ conoids were measured for each line (−IAA vs. +IAA). **b–d** Parasites grown in ±IAA for 40–44 h were collected and stimulated with 3 μM A23187 for 10 min prior to processing. **a**, **c** Representative of three or more experiments with similar outcomes. **b**, **d** Statistics were performed with one-way ANOVA with Tukey's multiple comparison

CPH1 was insoluble in detergent extracted cells (Supplementary Fig. 4a). Biotin labeling with BirA fused to CPH1 (Supplementary Fig. 4b) detected many additional protein bands that were detected with streptavidin compared to the parental control (Supplementary Fig. 4c). These additional biotinylated proteins were concentrated at the apical end, as shown by staining with fluorescently labeled streptavidin (Supplementary Fig. 4d). These results indicated that the BirA fusions target correctly and that this approach was amenable to identify proteins that interact with CPH1. To broaden this approach and build a proximity-based, protein–protein interactome of the conoid, we also constructed BirA fusions with other conoid proteins including the essential motor protein MyoH[11] and RNG2 (Supplementary Fig. 4e, f). We also included several calmodulin-like proteins that have recently been localized to the conoid and which may regulate MyoH (e.g., CaM1, CaM2, and CaM3)[23]. These BirA fusions all effectively

labeled the apical tip of the cell, in addition to showing strepta-vidin labeling of the apicoplast, which is due to endogenous biotinylated proteins (Supplementary Fig. 4d, g)[23]. Less intensive labeling at the apical end of the parasite was observed for BirA fused to RNG2, a result likely due to differences in the efficiency of labeling. Biotinylation followed by streptavidin capture and LC/MS/MS (LC, liquid chromatography; MS = mass spectrometry) was used to identify labeled proteins from the BirA fusion lines and the control parental line ku80[KO].

To identify potential interactions from these complex data sets, we used the program Straightforward Filtering IndeX program (SFINX), which provides a statistically robust method for determining true interactions while filtering out false-positives from replicate data sets[31]. SFINX also displays a network or interactome based on a sliding window of strictness that defines the cutoff for meaningful interactions. Here, we included

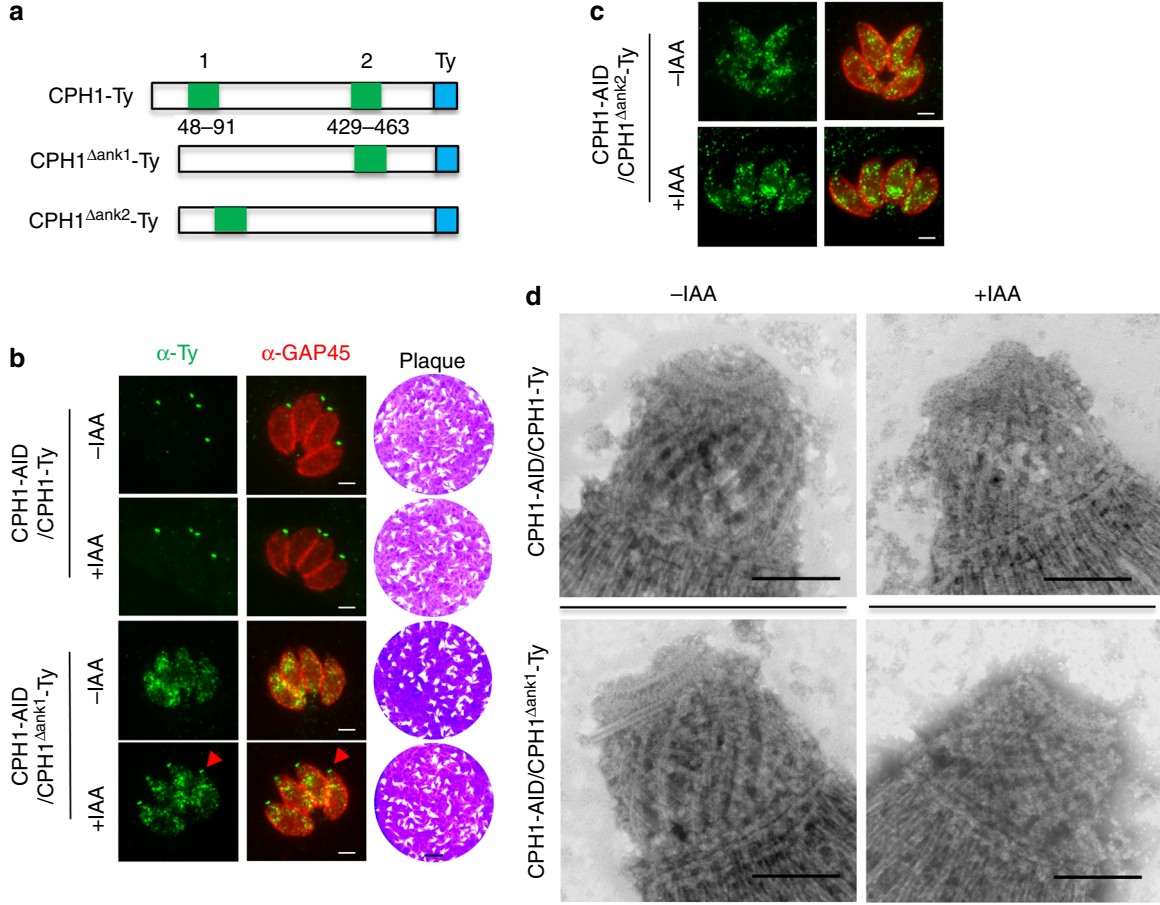

**Fig. 4** Ankyrin repeats target CPH1 to the conoid and are critical for function. **a** Model of CPH1 complementation constructs. The ankyrin repeat 1 (48–91) or ankyrin repeat 2 (429–463) were deleted from the wild-type copy CPH1-Ty to create copies of CPH1$^{\Delta ank1}$-Ty and CPH1$^{\Delta ank2}$-Ty, respectively. **b** Expression of stable complementing lines in the CPH1-AID line. Localization by IFA (left) following culture in ±IAA for 24 h, scale bar = 2 μm. Red arrows point to the partial conoid localization of CPH1$^{\Delta ank1}$-Ty. Plaque formation (right) following growth in ±IAA for 7 days (right). Scale bar = 0.5 cm. **c** Transient expression of CPH1$^{\Delta ank2}$-Ty, the CPH1-AID line. Localization as in **b**. **d** Conoid integrity in extracellular parasites examined by negative staining and TEM. Parasites were grown in ±IAA for 40–44 h, collected, and stimulated with 3 μM A23187 for 10 min to protrude the conoid. Scale bar = 250 nm. **b**–**d** Representative of three or more experiments with similar outcomes

the full interactome network with a strictness = 1 (e < 4 × 10$^{-5}$) (Fig. 5a; Supplementary Data 1 contains the input file for SFINX and Supplementary Data 2 contains a summary of the output). This analysis revealed 19 proteins that interact with CPH1 (Fig. 5a). Surprisingly, all of these partners were also identified using MyoH-BirA, and many of those were also shared with RNG2-BirA (Fig. 5a and Supplementary Data 2). Included in the common binding partners of CPH1, RNG2, and MyoH is the tubulin-binding protein DCX (Fig. 5a), which has been implicated in formation of the novel tubulin-rich fibers that comprise the conoid[18]. When the SFINX interactome was run without MyoH data sets, we identified several tubulins as being more enriched with CPH1-BirA (Supplementary Data 3), suggesting CPH1 interacts with the tubulin-rich conoid fibers. We performed labeling with RNG2 that was tagged with BirA at the N and C termini (i.e., BirA-RNG2 and RNG2-BirA). Although SFINX is not able to parse these data sets separately, when analyzed using the statistical package within Scaffold, we identified only two significant differences in the labeling pattern (Supplementary Table 6). Two hypothetical proteins were specifically labeled with BirA-RNG2: one of these contained a microtubule associated protein domain, while another showed no homology (Supplementary Table 6). In a previous study using BirA fusions of CaM1, CaM2, and CaM3, these

CaM-like proteins were found to label a number of proteins including MyoH[23]. When these data were included in the SFINX analysis here, CaM1 and CaM2 were not directly clustered within the MyoH interactome, but rather interact indirectly through common partners including IMC1 or MyoA (Fig. 5a, Supplementary Data 2). CaM1 and CaM2 also share a number of interaction partners (Fig. 5a, Supplementary Data 2), consistent with previous data that they are partially redundant and control the function of MyoH[23].

**CPH1 mediates important interactions with conoid proteins.** CPH1 forms a prominent node in the interactome that closely interacts with an overlapping set of proteins shared with MyoH and RNG2 (Fig. 5a). Overall, 22 of the 24 proteins found in the CPH1- MyoH-RNG2 interactome were also present in the original conoid proteome[8], indicating good concordance with previous data. To determine the localizations for all 24 proteins, we analyzed the remaining six uncharacterized proteins by CRISPR tagging technology, and found two of these six proteins also localized to the apical (Supplementary Fig. 6). In total, 14 of the 24 of the proteins in this interactome were localized to the apical complex, 3 were associated with microtubules, and 2 were found

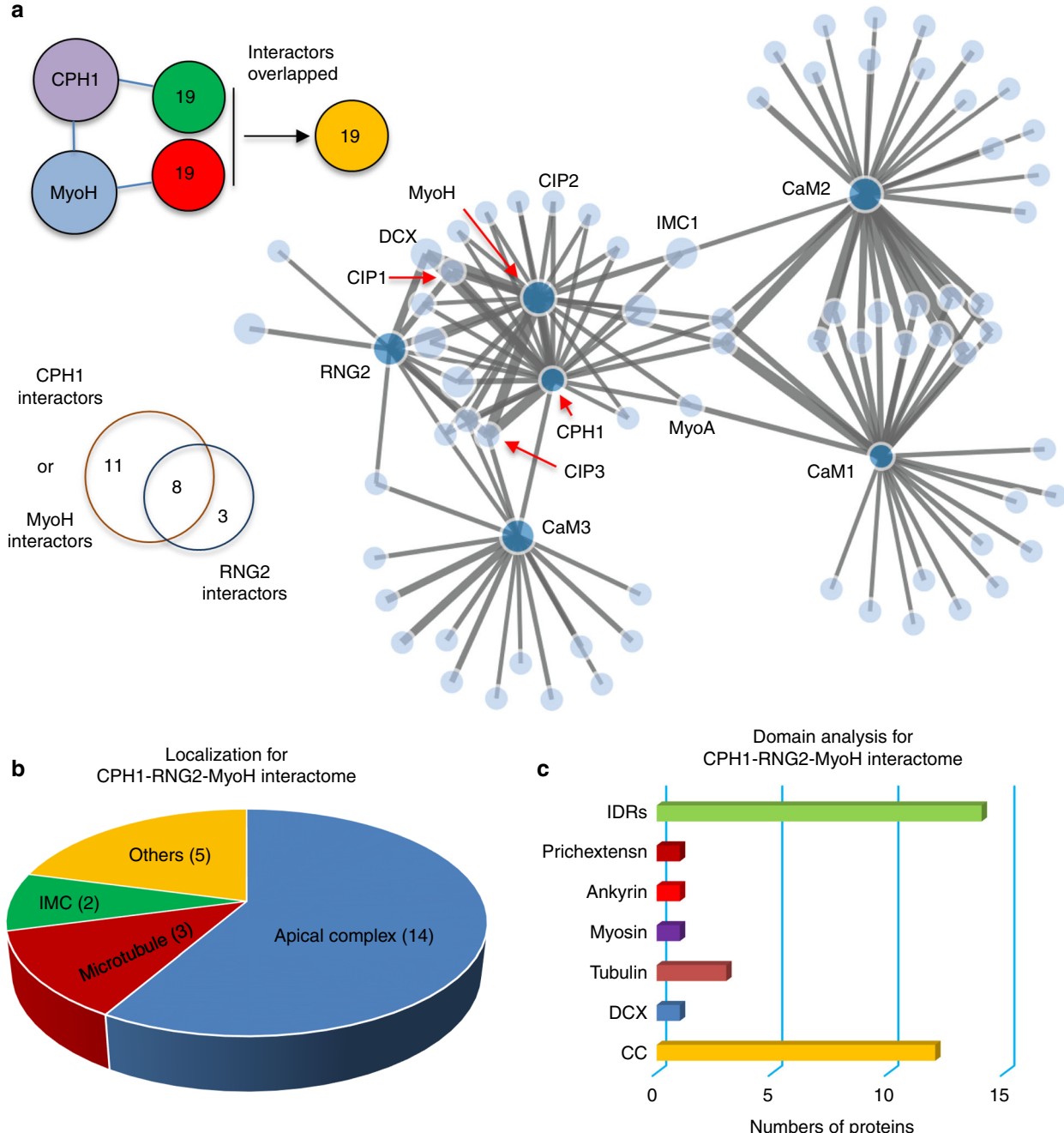

**Fig. 5** CPH1 is a central hub interacting with multiple proteins at the conoid. **a** Replicate mass spectrometry data sets generated with BirA fusion lines were analyzed using SFINX (http://sfinx.ugent.be/) to generate a proximity-based protein interaction network. Interactors of CPH1 and MyoH completely overlapped, and were partially shared with those for RNG2. Baits (BirA fusions)—deep blue nodes; prey (interactors)—light blue nodes. See Supplementary Data 2. **b** Localizations for the CPH1-MyoH-RNG2 interactome were analyzed for annotated and hypothetical proteins identified in the CPH1-MyoH-RNG2 interactome. Pie chart shows percentages of each category. The number in each category indicates protein number. $N = 24$ interactors. **c** Domains were analyzed for apical proteins and microtubule in the CPH1-MyoH-RNG2 interactome with InterPro. $N = 17$ interactors. **b**, **c** Protein localizations and domains for proteins in the CPH1-MyoH-RNG2 interactome were determined as defined in Supplementary Data 2. See also Supplementary Data 4, 5 and Supplementary Fig. 5 for GO term analysis. CC coiled coil, DCX double cortin microtubule-binding domain, myosin myosin A and H, prichextensin proline-rich motif present in plant cell wall, IDRs intrinsically disordered regions

in the IMC, while 5 were determined to other locations (Fig. 5b and Supplementary Data 2). These apically localized proteins contain various functional domains, as defined by InterPro domain analysis, including ankyrin repeats, myosin motor domains, coiled coil (CC) domains, and the tubulin-interacting domain called DCX (Fig. 5c). In addition to these conserved domains, many of the proteins in the interactome contain

intrinsically disordered regions (IDRs). Gene ontology term analysis for the eight annotated proteins in the interactome based on cellular compartment identified scaffolding-related structures, such as IMC, pellicle, and cytoskeleton: based on molecular function, they were classified as nucleotide binding, hydrolase, and motor activities (Supplementary Fig. 5, Supplementary Data 4, 5).

Six of the 11 apically localized hypothetical proteins that we initially localized in Fig. 1 were also present in the CPH1-MyoH-RNG2 interactome. However, other than CPH1, these proteins were not individually essential (Supplementary Fig. 2). Therefore, we examined the role of CPH1 in controlling the localization of the essential myosin motor MyoH and partially essential proteins DCX and RNG2 by tagging the genes with CRISPR in the CPH1-AID line (Fig. 6a). In intracellular parasites RNG2, DCX, and MyoH remained apically localized when CPH1-AID was degraded by addition of auxin (Fig. 6b). In extracellular parasites, MyoH was partially re-localized to the cytosol, while DCX and RNG2 remained unchanged when CPH1 was degraded by addition of auxin (Fig. 6c). However, their interactions with CPH1 were confirmed by proximity ligation assay (PLA) (Fig. 6d). These findings suggest that the profound defect of CPH1 deficient parasites is not readily explained by loss of localization of known apical proteins.

We next extended our analysis to examine several partners of CPH1 that are hypothetical unknowns and predicted to play important roles during in vitro growth based on having high CRISPR fitness scores[32]. We chose three hypothetical proteins that contained CC domains and IDRs, while one of them included a proline-rich extension ("prichextensn") domain (Fig. 7a). Again, CRISPR was used to tag the genes in the CPH1-AID line. All three hypothetical proteins were localized to the apical complex, and they were hereafter named CPH1-interacting proteins (CIP1, CIP2, and CIP3) (Fig. 7a). In intracellular parasites, CIP1, CIP2, and CIP3 remained apically localized when CPH1 was degraded by addition of auxin (Fig. 7a). The concentration of CIP1 and CIP3 remained constant, while CIP2 was slightly downregulated under these conditions (Fig. 7a). In extracellular parasites, CIP1 and CIP2 were entirely re-localized to the cytosol, while CIP3 remained apical when CPH1 was degraded by addition of auxin (Fig. 7a). The destabilization of these CIPs may result from conoid collapse upon depletion of CPH1; hence, their interactions with CPH1 were further examined by PLA, confirming their close proximity (Fig. 7b). When these *CIP* genes were individually deleted using CRISPR/Cas9 (Supplementary Fig. 7a), Δ*cip1* or Δ*cip2* showed no growth defect, while Δ*cip3* had a mild growth defect (Supplementary Fig. 8a, b). Though the double mutant Δ*cip1*Δ*cip2* had no growth defect, the triple Δ*cip1*Δ*cip2*Δ*cip3* mutant had profound defects in plaque number and size (Fig. 7c, Supplementary Fig. 7b and Supplementary Fig. 8a). The conoid architecture was examined by negative staining and EM (Fig. 7d), which demonstrated that Δ*cip1*Δ*cip2* was prone to loss of the pre-conoid ring, while Δ*cip1*Δ*cip2*Δ*cip3* had a high frequency of disrupted or collapsed conoids. Collectively, these findings indicate that the essential nature of CPH1 stems from its interaction with multiple proteins that are required for conoid stability.

## Discussion

*Toxoplasma gondii* has a highly developed apical complex, including a tubulin-rich conoid that is typical of coccidian members of the phylum. Although previous proteomic studies have identified a number of candidate conoid proteins, most of these are hypothetical unknowns, and very few have been studied functionally. We narrowed the list of putative conoid proteins by matching their expression profile to known apical complex proteins as localizing them the apical complex. We focused on one of these candidates, called CPH1, which contains two ankyrin repeats. Regulated degradation of CPH1 determined that it was essential for growth in vitro. Proximity-based biotinylation, using BirA fusions to key components of the apical complex, revealed that CPH1 is a protein–protein interaction hub that coordinates a number of important apical complex members. Collectively, these studies reveal that CPH1 forms a core component of the conoid, together with other rapidly evolving proteins, which comprise this unique MTOC.

The apical complex is largely known from ultrastructural studies that define common features of the phylum Apicomplexa[2]. The coccidian group of apicomplexans, including *T. gondii*, contains a tubulin-rich conoid, comprised of 14 fibers that form a spiral cone, flanked by pre-conoidal rings at the tip and the apical polar ring at the base[7]. Normally, alpha/beta tubulin subunits assemble into a closed microtubule, most commonly comprised of 13 protofilaments, and the singlet microtubules in *T. gondii* conform to this architecture[7]. The unique conformation of tubulin fibers in the conoid, which form non-fused fibers of nine protofilaments, is thought to be mediated by the tubulin-binding protein DCX[18]. In addition to tubulin, and its regulatory proteins, previous studies have localized a number of unique proteins to this structure including SAS6L, located in the pre-conoidal rings[19], and RNG2, the C-terminus of which is anchored at the apical polar ring while the N-terminus tracks with base of the conoid[20]. A previous proteomic study estimated that there may be ~250 proteins in the apical complex[8]. Here we narrowed this list to identify hypothetical proteins that match the cell cycle-dependent expression profile of known apical complex proteins[22]. We examined a subset of 21 conoid-enriched proteins, demonstrating that half of these proteins localized to the apical complex, while only 1 of these was essential for growth in vitro. This frequency is also consistent with the fitness defects seen in a genome-wide screen in *T. gondii* using CRISPR/Cas9[32] and in yeast, where essential proteins often function as interaction hubs[33]. Whether this reflects redundancy, or simply a role for many genes that is contextual (e.g., genes required in vivo but not in vitro), is uncertain. Nonetheless, our finding the CPH1, which is an essential conoidal protein, also functions as a protein hub. Following its degradation, parasite motility, invasion, and egress were all blocked, although it had no effect on microneme secretion, or cell replication. Instead, ultrastructural studies indicated that CPH1 was required for the stability of the conoid in extracellular parasites.

Similar to other apical complex proteins (e.g., MyoH, tubulins), CPH1 is insoluble in non-ionic detergents, precluding the use of pull-down experiments to identify potential binding partners. Instead, we turned to a proximity-based labeling system based on the permissive biotin ligase BirA[30]. This system has previously been used to identify components of the IMC[34] and exported proteins[35] in *T. gondii*. Here, we fused BirA to CPH1, as well as other apical complex proteins MyoH and RNG2. We also analyzed data from a recent study demonstrating that several calmodulin-like proteins (CaMs) are required for MyoH function[23]. The combined results of these BirA labeling experiments generated a large number of potential interactions for each of the bait protein. To decipher meaningful interactions among this set of candidates, we used a program called SFINX[31] to generate a collective interactome. These analyses reveal a core set of common partners for CPH1 and MyoH that mostly overlapped with RNG2. Among these was DCX, a tubulin-binding protein that is implicated in stability of the tubulin-rich fibers in the conoid[18]. These findings suggest that the motor protein MyoH may be tethered to the tubulin-rich conoid through these complex interactions. In contrast, the apically localized CaMs[23], defined peripheral nodes that lie outside the main interactome. Although the BirA fusions of CaMs detected MyoH, the reciprocal interaction is not prominently seen, suggesting these interactions are more transient. Alternatively, this finding may result from the C-terminal BirA fusion on MyoH being unable to access the small

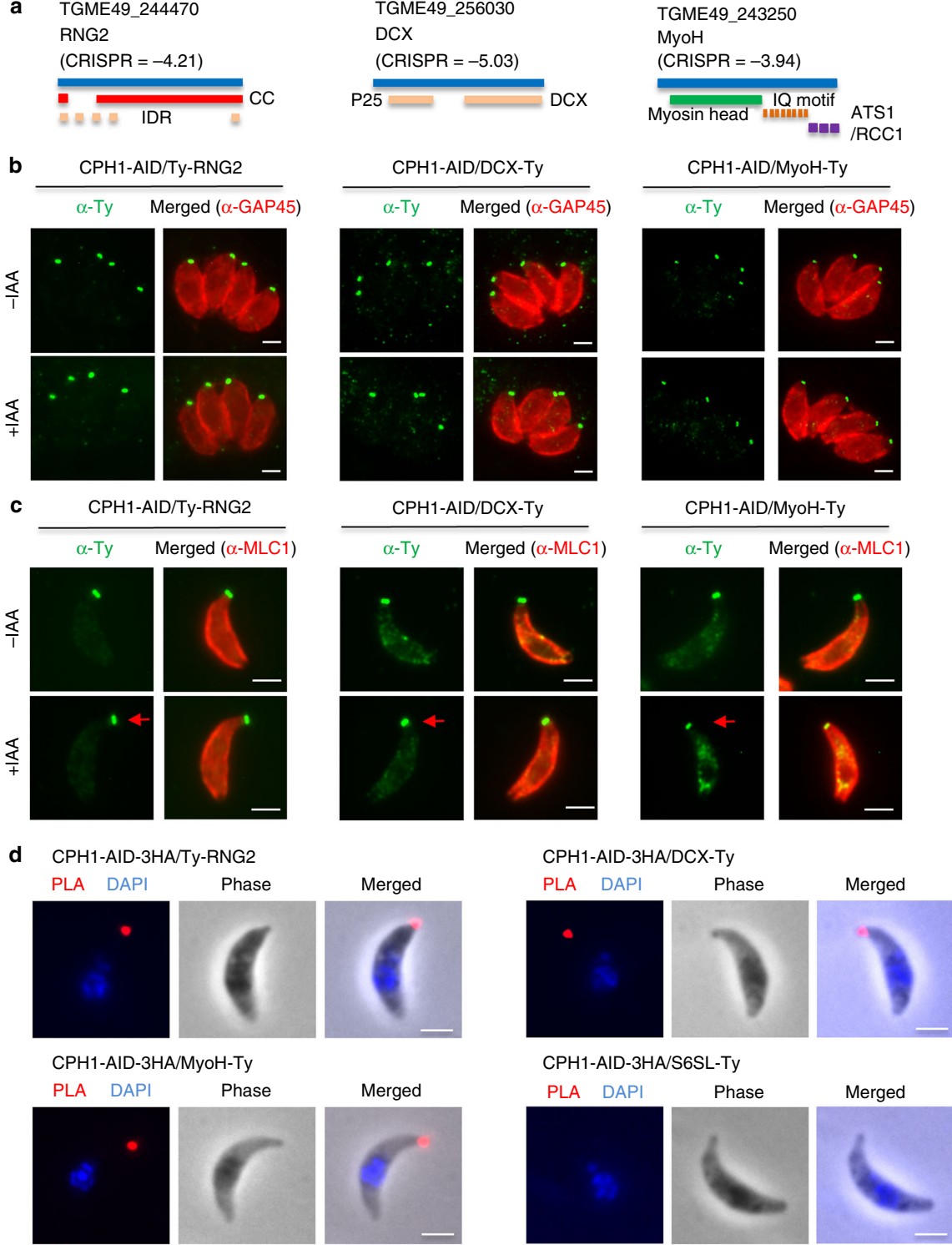

**Fig. 6** CPH1 depletion has a minimal effect on protein stability of known conoid proteins. **a** Domain arrangements of RNG2, DCX, and MyoH as determined by InterPro. CRISPR fitness scores were derived from ref. [32]. **b** Intracellular parasites grown in ±IAA for 24 h were stained with mouse anti-Ty (green) to detect the tagged proteins and rabbit anti-GAP45 (red) to outline the parasite followed by secondary antibodies conjugated to Alexa Fluor dyes. Scale = 2 μm. **c** Extracellular parasites collected after natural or mechanical egress from cultures grown in ±IAA for 40–44 h were stained with mouse anti-Ty (green) and rabbit anti-MLC1 (red) followed by secondary antibodies conjugated to Alexa Fluor dyes. Red arrow indicates position of the conoid. Scale = 2 μm. **d** Extracellular parasites were treated with 3 μM A23187 to protrude the conoid, and stained with rabbit anti-HA and mouse anti-Ty antibodies, and followed with reagents for proximity ligation assay (PLA). S6SL is a pre-conoid protein. Phase, DAPI (blue) and PLA (red) are shown. Scale = 2 μm. **b–d** Representative of three or more experiments with similar outcomes

CaM proteins that are predicted to bind within the internal IQ motifs of MyoH[23].

The proximity-based interactome is consistent with CPH1 functioning as a protein hub, which interacts with MyoH and a common set of partners. The phenotype of CPH1 depleted parasites mimics that of MyoH shutdown, or of the simultaneous depletion of CaM2 and degradation of CaM1, which are thought to regulate MyoH[23]. Loss of CPH1 only partially affected the distribution of MyoH in extracellular parasites, suggesting that the phenotype of CPH1 is due to other partners. CPH1 interacts with a number of proteins that have strong fitness defects in a genome-wide CRISPR screen[32]. Interestingly, the conoid forms normally in intracellular parasites that lack CPH1, and its key binding partners are stable and traffic correctly to the conoid. However, in extracellular parasite the conoid structure is adversely affected by loss of CPH1, and several of its core

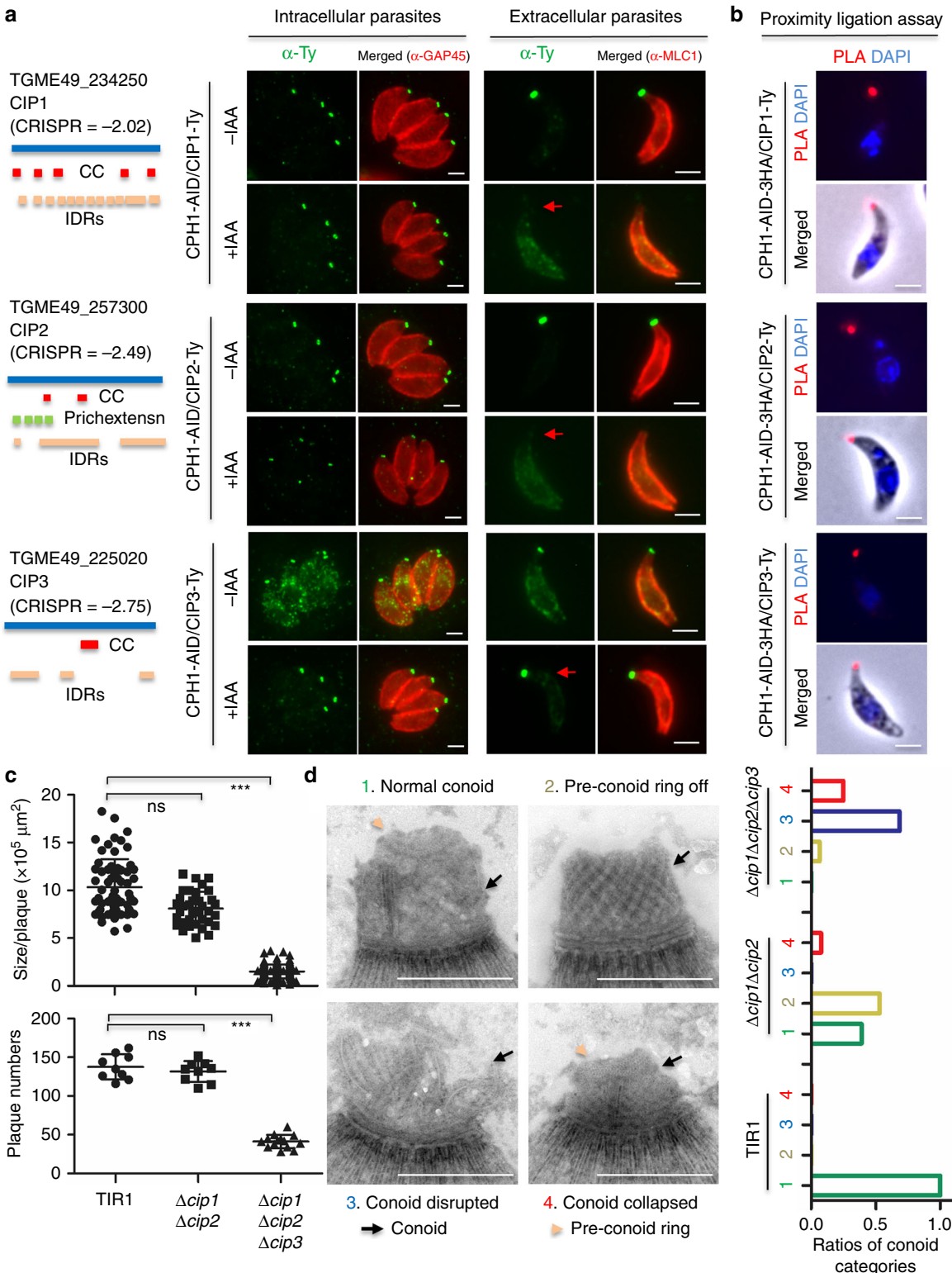

**c** Plaque assays showing Size/plaque (×10⁵ μm²) and Plaque numbers for TIR1, Δcip1Δcip2, and Δcip1Δcip2Δcip3.

**d**
1. Normal conoid
2. Pre-conoid ring off
3. Conoid disrupted — → Conoid
4. Conoid collapsed — ▶ Pre-conoid ring

Ratios of conoid categories

interaction partners were re-localized to the cytosol including CIP1, CIP2. In contrast, CIP3, DCX, and RNG2 were largely unaffected. The interaction of CIP proteins with CPH1 was further supported by PLA, which detects interactions of <30 nm[36]. This distance is similar to the original estimates of the labeling radius of BirA (20–30 nm)[37] although this distance was refined by later studies to an effective radius of 10 nm[38]. Although CIP1, CIP2, and CIP3 were individually dispensable, their triple deletion led to a profound defect in the conoid stability. These findings suggest that the pronounced defect with loss of CPH1 is due to loss of hypothetical unknowns, including CIP1, CIP2, and CIP3, while core structural components of the conoid like DCX and the microtubule-rich fibers were less affected.

CPH1 is highly conserved in apicomplexans including coccidians, which contain a well-developed conoid, and the more deeply branching organism *Cryptosporidium*. Despite conservation of the gene, the ankyrin repeat domains are less well conserved and not readily identifiable outside of the more closely related tissue-cyst coccidians (i.e., *Toxoplasma*, *Neospora*, and *Hammondia*). This pattern suggests that CPH1-binding partners and functions may vary between different organisms in the phylum. CPH1 is also conserved in organisms that lack a tubulin-rich conoid, but instead have an apical polar ring, such as *Plasmodium* spp. In this regard, CPH1 may be similar to SAS6L, which is located in the pre-conoidal rings in *T. gondii*[39] but in the apical complex in *Plasmodium berghei* where it is expressed in sporozoites and ookinetes[15]. Similarly the ortholog of CPH1 in *P. falciparum* is highly expressed in ookinetes and sporozoites (http://PlasmoDB.org). Given the essentiality of CPH1 in *T. gondii*, further study of orthologs among this group may further inform us about its broader role in the apical complex.

The phenotype of CPH1 is consistent with it acting as a protein–protein interaction hub, a term coined for nodes that have a large number of interacting partners and hence are often essential[40]. Their essentiality has been attributed to the fact that they interact with, and hence are likely to affect, a large number of proteins, hence increasing the likelihood that their loss will affect a critical function(s)[33]. One of the features of CPH1 that predicts it might act as a hub is the presence of two ankyrin repeats domains. The ankyrin repeat is a universal module of about 33 amino acid that forms a L-shaped structure consisting of two alpha-helices following a beta-hairpin loop[41]. Ankyrin repeat is a universal scaffold mediating protein–protein interactions by the beta-hairpin loop region and inner short helices[42]. Additionally, many of the interaction partners of CPH1 contain protein interaction domains including CC domains and IDRs. Coiled coil domain proteins often act as scaffolds and can contribute to formation of signaling networks[43]. IDR are highly flexible, they can adopt multiple conformations and often have multiple binding partners[44]. Coiled coil domains and IDRs evolve rapidly and have been associated with the functional expansion of the centrosome, another form of MTOC, in eukaryotic cells[43]. Our findings extend this paradigm to demonstrate that rapid evolution of protein interaction networks may also underlie the uniquely specialized biology of MTOCs in apicomplexan parasites.

## Methods

**Generation and cultivation of *Toxoplasma gondii* lines**. Parasite lines generated here (Supplementary Table 2) were grown in human foreskin fibroblasts (HFFs), originally obtained from the laboratory of John C. Boothroyd, Stanford Univ., collected and maintained mycoplasma negative, as described previously[25]. The RH$\Delta ku80\Delta hxgprt$ line[45], referred to as ku80KO, was used for CRISPR-mediated gene deletion and epitope tagging, as described previously[23,24] (Supplementary Tables 2–4). Auxin-induced degron (AID) fusions were generated in the TIR1-expressing line, described previously[26]. Degradation of AID-fusion proteins was induced with 500 µM auxin (IAA) or 0.1% ethanol alone. Antibodies used in this study are defined in Supplementary Table 5.

**Parasite replication, host cell invasion, and microneme secretion**. Replication was assessed by the number of parasites per vacuole at 24 h after invasion, as described previously[23]. Plaque formation on HFF monolayers was monitored by crystal violet staining of monolayers after 7 days of in vitro growth, and measurement of plaque sizes was performed, as described previously[23]. Parasite invasion into HFF cells was examined after 20 min challenge using a differential staining assay to discriminate invaded and attached parasites, as described previously[23]. Microneme secretion was monitored using the previously characterized MIC2-GLuc line for extracellular parasites that were stimulated with ±1% BSA and ±1% ethanol, as described previously[28].

**Light microscopy**. Immunofluorescence microscopy and super-resolution confocal microscopy on fixed cells were performed as previously described[23]. Antibody sources are listed in Supplementary Table 5. Conoid extrusion in response to calcium ionophore was monitored by phase contrast microscopy following stimulation with 3 µM A23187 for 10 min, as described previously[23]. Parasite motility was performed with extracellular parasites using video microscopy, as previously described[46]. Video microscopy was also used to monitor parasite egress and diffusion of soluble DsRed from the parasitophorous vacuole, as described previously[27]. Parasites expressing secretory DsRed[29] were grown in HFF on glass bottom dishes (MatTek) and imaged in Ringer's media containing 3% FBS using an Zeiss Axio Observer microscope equipped with 37 °C temperature and 5% $CO_2$. Alternating bright field and fluorescence images were acquired using a spinning disc confocal system with a ×40 oil Plan-Apochromat objective (N.A. 1.3, Ph3) and Evolve 512 Delta EMCCD camera (Photometrics), at a rate of ~1 frame per 5 s for a 15 min total period. Following stimulation with 2 µM A23187, diffusion of DsRed from the parasitophorous vacuole was recorded. Images were analyzed in ImageJ to monitor fluorescence intensity from eight separate vacuoles (intensity tracked in regions of interest over time) for each of three independent experiments. Proximity ligation assay was performed according to the manufacturer's protocol using the Duolink in situ PLA probes and red detection reagents (Sigma, DUO92101). Antibody sources are listed in Supplementary Table 5.

**Transmission electron microscopy**. For standard TEM, infected HFF cells were fixed in 1% glutaraldehyde (Polysciences Inc., Warrington, PA) and 1% osmium tetroxide (Polysciences Inc.) in 50 mM phosphate buffer at 4 °C for 30 min. Samples were then processed and sectioned, as described previously[47]. Sections were stained with uranyl acetate and lead citrate, and viewed on a JEOL 1200 EX transmission electron microscope (JEOL USA Inc., Peabody, MA).

**Negative staining and immunostaining for electron microscopy**. Fleshly harvested parasites were stimulated to protrude the conoid by treatment with 3 µM A23187 in PBS for 10 min, followed by extraction with 5 mM deoxycholate at room temperature for 10 min. The detergent insoluble cytoskeleton was collected by centrifugation at 800×*g* for 10 min, resuspended in distilled water, and allowed to absorb onto formvar/carbon-coated nickel grids for 10 min. Grids were stained with 1%

**Fig. 7** CPH1 mediates important interaction with hypothetical unknown conoid proteins. **a** Analyses of interactions with CIP1, CIP2, and CIP3. Left panel: domain analysis as determined by InterPro, and domain names as shown in Fig. 5c. CRISPR fitness scores were derived from ref.[32]. Right panels: intracellular and extracellular parasites were treated, fixed, and stained as described in Fig. 6b, c. Scale = 2 µm. **b** PLA for interaction of CPH1 with the CIPs was performed as described in Fig. 6d. Scale = 2 µm. **a**, **b** Representative of three or more experiments with similar outcomes. **c** *CIP1*, *CIP2*, and *CIP3* were sequentially deleted, generating double and triple knockouts. Plaque sizes and numbers are plotted (N > 50 plaques). Mean ± SEM for 3 experiments with 3 replicates for each (n = 9). ***P < 0.0001, Kruskal–Wallis test with Dunn's correction for multiple comparison. For details on generation of the parasite lines, see Supplementary Fig. 7. **d** Parasites were treated with 3 µM A23187 to protrude the conoid, prior to EM processing. Four major categories of conoid structure were identified with examples shown on left: (1) normal conoid from TIR1 parental line; (2) loss of the pre-conoid ring from Δ*cip1*Δ*cip2*; (3) disrupted conoid, and (4) collapsed conoid from Δ*cip1*Δ*cip2*Δ*cip3* (colored for the category numbers). Ratios of different categories were calculated from two independent experiments, and averages were shown (N = 20 conoids for each group). Scale = 500 nm. ns not significant

aqueous phosphotungstic acid, pH 7, (Electron Microscopy Services, Fort Washington PA) for 30 s, air-dried, and used for imaging. For immunolabeling, samples were blocked with 1% FBS for 5 min, and then incubated with mouse anti-Ty antibody (1:1000) or rabbit anti-HA antibody (1:250) for 30 min, followed by secondary goat anti-mouse/rabbit IgG antibody conjugated to 18 nm colloidal gold (Jackson ImmunoResearch Laboratories, Inc., West Grove, PA) for 30 min. Antibody sources are listed in Supplementary Table 5. Grids were then washed and stained with phosphotungstic acid as described above. Samples were viewed on a JEOL 1200EX transmission electron microscope (JEOL USA) equipped with an AMT 8 mega-pixel digital camera (Advanced Microscopy Techniques, Woburn, MA).

**Bioinformatic analysis**. Apicomplexan orthologs were identified by searching proteomes in HMMER (http://www.ebi.ac.uk/Tools/hmmer/) with *T. gondii* proteins as queries, and hits with $E$-value < $1e-7$ were retained. Phylogeny analyses were performed with Pylogeny.fr (http://www.phylogeny.fr/index.cgi) using default settings[48]. Protein alignment was constructed using MUSCLE and curated by Gblocks, the phylogeny was inferred with PhyML and aLRT, and the tree generated using TryDyn. Protein domains were analyzed with InterPro (https://www.ebi.ac.uk/interpro/). Classification of GO terms was performed in TOXODB (http://toxoDB.org).

**Mass spectrometry and interactome analysis**. Purified biotinylated samples on streptavidin beads were subjected to mass spectrometry analyses, as described previously[23]. Two biological replicates for each BirA line were analyzed, with a control untagged line (i.e., RHku80$^{ko}$) run in parallel for each round. Resulting spectra were searched against a combined database of *T. gondii* (http://ToxoDB.org, release 29) and human proteins and a decoy database, using Mascot and Scaffold. The data sets in Scaffold (v4.7.2) were filtered with min #peptide = 2, protein threshold ≤99%, and peptide threshold ≤95%. The resulting hit lists were exported to excel and edited to match the format with the SFINX analysis tool (http://sfinx.ugent.be) (Supplementary Data 1). A separate excel file containing BirA baits was uploaded to software SFINX website (http://sfinx.ugent.be/) together with the mass spectrometry data file. Figures were exported in png format.

**Statistical analyses**. One-way or two-way analysis of variance tests with Tukey's multiple comparison were performed to assess differences for data that were normally distributed. Kruskal–Wallis test with Dunn's multiple comparison was applied to analyze data that were not normally distributed or where sample sizes were too small to determine the distribution. $P \leq 0.05$ was considered significant; $0.0001 > P < 0.05$ values were reported precisely, while smaller values are indicated as $P \leq 0.0001$.

**Data availability**. The data supporting the findings of the study are available in this article and its Supplementary Information files, or from the corresponding author upon request. The proteomics data used for SFINX analysis has also been submitted to BioGRID (https://thebiogrid.org/).

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

## Acknowledgements

We are grateful to Jennifer Barks for assistance with cell culture, Dr. Wandy Beatty, Microbiology Imaging Facility for electron microscopy, Dr. Michael Naldrett at the Proteomics and Metabolomics Facility, Center for Biotechnology at the University of Nebraska-Lincoln for the proteomics analysis, and Dr. Kevin Titeca for assistance with SFINX graphics. Regents were kindly provided by Ke Hu (pminCre plasmid), Dominique Soldati (GAP45 and MLC1 antibodies), and Vern Carruthers (pDsRed plasmid). Supported by the National Institutes of Health, National Institute of Allergy and Infectious Diseases grant AI034036 (L.D.S). L.L.D. was supported by a grant from the National Science Foundation (DGE-1143954).

## Author contributions

S.L. conceived the study, designed, and performed the majority of experiments, analyzed the data, generated the figures, and wrote the manuscript. B.A. performed the super-resolution microscopy and electron microscopy. L.L.D. performed the video microscopy measurements and analyzed the data. L.D.S supervised the study, assisted with experimental design and analyses, and together with S.L. wrote the manuscript.

## Additional information

**Competing interests:** The authors declare no competing financial interests.

