## [Peer Review File · Nature Communications]

Reviewers' comments:

Reviewer #1 (Remarks to the Author):

This manuscript verifies 11 components of the *T. gondii* apical complex (which includes the tubulin-rich conoid) and investigates their functions. In particular, the authors identify an essential apical component, conoid protein hub 1 (CPH1; TGME49_266630).

The authors used Super-Resolution microscopy to show that CPH1 is concentrated in the middle part of the conoid, slightly anterior to RNG2 and slightly posterior to CaM1. Because of the possibility of chromatic aberration, it would be useful to reverse the fluorophores on the secondary antibodies to confirm that these very small differences in location are real.

This is particularly important as the MyoH-BirA and RNG2-BirA proximity partners overlapped extensively with the CPH1-BirA proximity partners. Assuming the different spatial location is confirmed, the overlap of the datasets indicates that the BirA system is labelling proteins somewhat indiscriminantly. This calls into question the authors claim that CPH1 acts as a protein interaction hub. Evidence for direct interactions between the interacting partners is limited. It remains possible that they may simply be located in the same region of the cell. An alternative method such as the proximity ligation assay would help strengthen the claim that the reported interacting proteins really are closely located.

The interacting partner analysis involves appropriate controls eg, comparison of biotinylated proteins in parent untagged RHku80KO and CPH1-BirA lines. It is unclear why the CPH-1 and MyoH-BirA lines have intense biotin signal at the conoid but the RNG-2 lines have less biotin labelling (by IFA) in panel S3 D versus G. Please comment.

The authors report that auxin-inducible degron knock-down of CPH1-AID reduced conoid length, rendered parasites immobile and prevented plaque formation – a very interesting finding.

The authors further showed that CPH1 with a deletion in ankyrin repeat region 1 (CPH1 Δ Ank1-Ty) is mis-targeted and does not reach the conoid in the absence of auxin but is partially re-localized to the apical end, and able to complement the CPH1-AID line growth in the presence of auxin. The authors were not able to generate a stable line of CPH1 Δ Ank2-Ty and suggest a dominant negative effect.

The authors conclude that the ankyrin repeat regions target CPH1 to the conoid. However, it is also possible that the ankyrin repeat regions are needed for correct folding of CPH1 and that the binding domain is in another region of the protein. This possibility should be discussed.

Line 274. The authors state that 12 of the 24 proteins in the CPH1-MyoH-RNG2 interactome were localized to the apical complex, 3 were associated with microtubules, and 2 were found in the IMC, while others were not determined or found in other locations. Only summary data is presented to support this statement. The original images should be provided.

The authors also characterised three proteins, which they refer to as CPH1-interacting (CIP1-3), whose location is disrupted upon CPH1 knock-down. Disruption of CIP1 or CIP2 is individually not lethal but the double disruption exhibits synthetic lethality. It would be useful to know if the conoid architecture is altered in this Δ cip1 / CIP2-AID knock-down.

The functions of CPH1, CIP1 and CIP2 remain unclear, other than the broad assumption that they are "structural proteins". Nonetheless, this study adds significantly to our understanding of the key players in conoid function in *T. gondii*.

Minor points.

Line 24 "includes spiral cap" should be "includes a spiral cap"

Line 115 - Of the 21 proteins examined, 11 were verified apical, 1 was found to be partly apical, leaving 9 that are in other compartments.

S1 Fig, Line 11. It is not clear on what basis 74 was localized to the cortical cytoskeleton and 52 was determined to be mitochondrial.

Reviewer #2 (Remarks to the Author):

The manuscript by Long and co-workers investigates a specialized structural compartment of the apicomplexan parasite *Toxoplasma gondii*, termed "conoid". The authors used CRISPR/CAS mediated tagging of 21 putative conoid associated proteins, identified by a *in silico* search using published transcription data. 11 of those candidates showed the expected localization and were functionally analyzed by gene knock-out. Only 1 protein was shown to be refractory to the CRISPR mediated KO strategy and termed "Conoid Protein Hub 1" (CPH1). CPH1 was analyzed in depth by high-resolution microscopy and inducible knock-down strategies which showed that CPH1 is involved in conoid stability and has an impact on motility and host cell invasion. Using BirA based proximity labeling of CPH1 they described a CPH1 centered protein network of 19 proteins. Combined with MyoH-BirA and RNG2-BirA based experimentally identified interactomes the authors present 24 interacting proteins. Again, they localize these proteins and show that 12 of these have an apical staining pattern.

In summary, this manuscript provides important insights into the unique cell biology of this single cell organism. It is using state-of-the art approaches, is well written and experimentally straightforward. It leaves me only with minor points that should be addressed prior to publication.

1. 109-117: Please provide search parameters for the data mining of "... candidates with similar expression profile".
2. Line 143-147: The authors should consider to move Figure 1E to Figure 4, as the function of the ankyrin repeats are studied there.
3. Line 160: The authors state that treatment of the CPH1-AID line with auxin did not affect replication during 24 hr showing a normally developed parasite in Figure 2C as evidence. This finding should be corroborated by quantification.
4. Line 261-262: Please change reference style "In previous studies using BirA fusions ...were found to label a number of proteins including MyoA (Long, 2017)".
5. Line 272: The authors state that they localized 24 proteins of the defined interactome assumingly again by CRISPR mediated HA tagging. Please state this in the text.
6. Fig. 5B: Please insert the numbers of proteins that show the described localization phenotype. How did the authors prioritize the proteins given that some were not tagged?

Several minor language and spelling mistakes should be corrected throughout the manuscript. A few examples:

125: ...was refractory to CRISPR/Cas9 deletion...

372: Among these was DCX, a tubulin binding protein that is implicated in stability the tubulin-rich fibers in the conoid...

717: phrase contrast

Reviewers' comments:

Reviewer #1 (Remarks to the Author):

This manuscript verifies 11 components of the *T. gondii* apical complex (which includes the tubulin-rich conoid) and investigates their functions. In particular, the authors identify an essential apical component, conoid protein hub 1 (CPH1; TGME49_266630).

1) The authors used Super-Resolution microscopy to show that CPH1 is concentrated in the middle part of the conoid, slightly anterior to RNG2 and slightly posterior to CaM1. Because of the possibility of chromatic aberration, it would be useful to reverse the fluorophores on the secondary antibodies to confirm that these very small differences in location are real.

Response: The reviewer raises an important point. However, we did not intend to claim that this small difference was biologically meaningful. Rather it is evident from the SR microscopy that the majority of the signals overlap in a broad region. This is also demonstrated by the immunoEM experiments, where the region of immunogold staining overlaps for the different proteins. Hence, we do not think there would be much gained by repeating the experiments with inverted fluorophores. Instead, we have modified the text so as to not overstate the importance of this minor difference.

2) This is particularly important as the MyoH-BirA and RNG2-BirA proximity partners overlapped extensively with the CPH1-BirA proximity partners. Assuming the different spatial location is confirmed, the overlap of the datasets indicates that the BirA system is labelling proteins somewhat indiscriminantly. This calls into question the authors claim that CPH1 acts as a protein interaction hub. Evidence for direct interactions between the interacting partners is limited. It remains possible that they may simply be located in the same region of the cell. An alternative method such as the proximity ligation assay would help strengthen the claim that the reported interacting proteins really are closely located.

Response: We agree with the suggestion by the reviewer that it is important to confirm the interactions suggested by BirA labeling using another method. We used proximity ligation assay (PLA) to confirm the protein-protein interactions. Previous studies suggested that PLA works for protein interactions with distance of < 30 nm (Soderberg O, et al. 2015), depending on the length of oligonucleotide conjugated to the probes. We used PLA to examine the interaction of CPH1 with N-terminus of RNG2, DCX, MyoH, CIP1, CIP2 and CIP3. These findings are shown in Fig 6,7 and they confirm the proximity labeling studies performed using BirA. Previous studies suggested that the BirA labels proteins at a radius of 20-30 nm (Van Itallie CM, et al. 2013) with a practical labeling radius later refined to be 10 nm (Kim DI et al. 2014). Based on these combined data, we believe that the BirA labeling system can serve as a system to identify interacting proteins.

References:

Van Itallie CM, et al. 2013. The N and C termini of ZO-1 are surrounded by distinct proteins and functional protein networks. *JBC*. 288(19):13775-13788.

Kim D.I, Birendra K.C, Zhu W, Motamedchaboki K, Doye K and Roux KJ. 2014. Probing nuclear pore complex architecture with proximity-dependent biotinylation. *PNAS*, 111(24):E2453-61.

Soderberg, O., Gullberg, M., Jarvius, M., Ridderstrale, K., Leuchowius, K. J., Jarvius, J., Wester, K., Hydbring, P., Bahram, F., Larsson, L. G. and Landegren, U. (2006). Direct observation of individual endogenous protein

complexes *in situ* by proximity ligation. *Nat Methods* 3(12): 995-1000.

3) The interacting partner analysis involves appropriate controls eg, comparison of biotinylated proteins in parent untagged RHku80KO and CPH1-BirA lines. It is unclear why the CPH-1 and MyoH-BirA lines have intense biotin signal at the conoid but the RNG-2 lines have less biotin labelling (by IFA) in panel S3 D versus G. Please comment.

Response: The reviewer raises an important point and we have addressed this in the revised text. The activity the BirA tag may be affected by fusion to different bait proteins, or by fusion at the C-terminus and N-terminus. Additionally, the expression level of different proteins may be different, which can also affect BirA activity.

The authors report that auxin-inducible degron knock-down of CPH1-AID reduced conoid length, rendered parasites immobile and prevented plaque formation – a very interesting finding.

The authors further showed that CPH1 with a deletion in ankyrin repeat region 1 (CPH1 Δ Ank1-Ty) is mis-targeted and does not reach the conoid in the absence of auxin but is partially re-localized to the apical end, and able to complement the CPH1-AID line growth in the presence of auxin. The authors were not able to generate a stable line of CPH1 Δ Ank2-Ty and suggest a dominant negative effect.

The authors conclude that the ankyrin repeat regions target CPH1 to the conoid. However, it is also possible that the ankyrin repeat regions are needed for correct folding of CPH1 and that the binding domain is in another region of the protein. This possibility should be discussed.

Response: It is commonly believed that ankyrin repeats play a role on mediating protein-protein interaction, by serving as a protein interaction architecture (Mosavi LK et al., 2004). However, it is also possible, as suggested by the reviewer, that the ankyrin repeats are important for correct folding of CPH1 while the binding domain is in another region. The fact that the conserved presence of CPH1 in Apicomplexa did not co-evolve with the ankyrin repeats in the protein, suggests that the ankyrin repeats in CPH1 may have different roles other than binding with partners. This possibility has been discussed in the revision.

Mosavi LK, Cammett, TL., Desrosiers, DC. & Peng, ZY. 2004. The ankyrin repeat as molecular architecture for protein recognition. *Protein Sci* 13, 1435-1448.

Line 274. The authors state that 12 of the 24 proteins in the CPH1-MyoH-RNG2 interactome were localized to the apical complex, 3 were associated with microtubules, and 2 were found in the IMC, while others were not determined or found in other locations. Only summary data is presented to support this statement. The original images should be provided.

Response: We have tagged and localized the 6 remaining uncharacterized proteins, and presented the results in S6 Fig. We have also revised the S6 Table to include localization information and identify experimental results by listing individual figures or supplemental figures for each of the partners in the interactome.

The authors also characterised three proteins, which they refer to as CPH1-interacting (CIP1-3), whose location is disrupted upon CHp1 knock-down. Disruption of CIP1 or CIP2 is individually not lethal but the double disruption exhibits synthetic lethality. It would be useful to know if the conoid architecture is altered in this Δ cip1 / CIP2-AID knock-down.

Response: We tested the essentiality of *CIP1* and *CIP2* and found that a double deletion mutant of these two genes was viable. This surprising result led us to reassess our original finding of the auxin induced degradation of the $\Delta cip1$ / CIP2-AID knock-down. The defect in this clone proved to be an artifact of using too high of a concentration of auxin that led to an ethanol-induced artifact. Importantly, this result only occurred in a limited number of experiments and is not a general feature of the auxin system, nor did it influence any of the other conclusions in the study. As a consequence, we generated a single mutants of $\Delta cip3$ and found that it had a mild defect in plaque formation (S7 and S8 Fig). Furthermore, although the double knockout $\Delta cip1\Delta cip2$ had no obvious defect, the triple knockout $\Delta cip1\Delta cip2\Delta cip3$ had profound defects in plaque number and size (Fig 7C, S8A Fig). We extended these studies to show that the triple knockout $\Delta cip1\Delta cip2\Delta cip3$ had severe defects in conoid architecture (Fig 7D). Collectively, these findings indicate that CHP1 plays an important role in organizing a number of proteins that control the structural stability of the conoid, including CIP1, CIP2 and CIP3.

The functions of CHP1, CIP1 and CIP2 remain unclear, other than the broad assumption that they are “structural proteins”. Nonetheless, this study adds significantly to our understanding of the key players in conoid function in *T. gondii*.

Minor points.

Line 24 “includes spiral cap” should be “includes a spiral cap”

Response: corrected.

Line 115 - Of the 21 proteins examined, 11 were verified apical, 1 was found to be partly apical, leaving 9 that are in other compartments.

Response: Corrected.

S1 Fig, Line 11. It is not clear on what basis 74 was localized to the cortical cytoskeleton and 52 was determined to be mitochondrial.

Response: We have removed the statements, as we agree that we cannot be specific about these localizations at the resolution provided here. However, as this is not the focus of the paper, we feel it is better to leave this question for a future study.

Reviewer #2 (Remarks to the Author):

The manuscript by Long and co-workers investigates a specialized structural compartment of the apicomplexan parasite *Toxoplasma gondii*, termed “conoid”. The authors used CRISPR/CAS mediated tagging of 21 putative conoid associated proteins, identified by a in silico search using published transcription data. 11 of those candidates showed the expected localization and were functionally analyzed by gene knock-out. Only 1 protein was shown to be refractory to the CRISPR mediated KO strategy and termed “Conoid Protein Hub 1” (CPH1). CPH1 was analyzed in depth by high-resolution microscopy and inducible knock-down strategies which showed that CHP1 is involved in conoid stability and has an impact on motility and host cell invasion. Using BirA based proximity labeling of CPH1 they described a CPH1 centered protein network of 19 proteins. Combined with MyoH-BirA and RNG2-BirA based experimentally identified interactomes the authors present 24 interacting proteins. Again, they localize these proteins and show that 12 of these have an apical staining pattern.

In summary, this manuscript provides important insights into the unique cell biology of this single cell organism. It is using state-of-the art approaches, is well written and experimentally straightforward. It leaves me only with minor points that should be addressed prior to publication.

1. 109-117: Please provide search parameters for the data mining of "... candidates with similar expression profile".

Response: We first identified the expression profiles for all the known proteins that are localized to the apical complex. We then manually searched and compared all the expression profiles of proteins identified in a previous proteomic study. This process allowed us to identify similar profiles for the new genes. The pattern of these genes is that expression reaches peak level during S to M phase, but drops immediately during cytokinesis (C), and drops to the lowest level at G1 phase. These features are described in the results.

2. Line 143-147: The authors should consider to move Figure 1E to Figure 4, as the function of the ankyrin repeats are studied there.

Response: We thank the reviewers for the suggestion. However, after considering this option, we think that the features of CPH1 should be discussed at the outset, as this is relevant to the evolutionary context of this protein. Hence, we have left the panel in figure 1E.

3. Line 160: The authors state that treatment of the CPH1-AID line with auxin did not affect replication during 24 hr showing a normally developed parasite in Figure 2C as evidence. This finding should be corroborated by quantification.

Response: We now present the results of replication in S3 Fig. There is a significant increase in the number of vacuoles containing 16 parasites and decrease in the number containing 1, likely is a result of a defect of parasite egress during the 24 hr culturing period.

4. Line 261-262: Please change reference style "In previous studies using BirA fusions ...were found to label a number of proteins including MyoA (Long, 2017)".

Response: Corrected.

5. Line 272: The authors state that they localized 24 proteins of the defined interactome assumingly again by CRISPR mediated HA tagging. Please state this in the text.

Response: The information of how the lines were generated (CRISPR tagging technology) was added in the text. In table S6, the localization information for each of the partners pointed to individual figures or pmid for reference.

6. Fig. 5B: Please insert the numbers of proteins that show the described localization phenotype. How did the authors prioritize the proteins given that some were not tagged?

Response: We have now localized all of the proteins and have indicated the numbers for each localization category in Fig 5 B and included the details in Table S6. We selected proteins to focus on based on essentiality (some were previously known) or high CRISPR fitness scores from a previous genome wide study. These criteria are defined in the results in the section describing Fig. 7.

Several minor language and spelling mistakes should be corrected throughout the manuscript. A few examples:

Response: The manuscript has been carefully re-examined on and we have corrected the items below and additional typos.

125: ...was refractory to CRSIPR/Cas9 deletion...

372: Among these was DCX, a tubulin binding protein that is implicated in stability the tubulin-rich fibers in the conoid...

717: phrase contrast

Reviewers' comments:

Reviewer #1 (Remarks to the Author):

"Conserved ankyrin repeat containing protein regulates conoid stability, motility, and cell invasion in *Toxoplasma*" by Dr Sibley and colleagues, Nature Communications.

The authors have answered most of the queries, however their response to the question about the CIP2-AID knock-down raises further questions.

The authors note that: "We tested the essentiality of CIP1 and CIP2 and found that a double deletion mutant of these two genes was viable. This surprising result led us to reassess our original finding of the auxin induced degradation of the Δ cip1 / CIP2-AID knock-down. The defect in this clone proved to be an artifact of using too high of a concentration of auxin that led to an ethanol-induced artifact."

The authors note that "...this result only occurred in a limited number of experiments and is not a general feature of the auxin system, nor did it influence any of the other conclusions in the study."

The authors refer back to their initial report using the auxin system in Toxo (which b.t.w is referenced incorrectly - the title of the 2017 paper is "Plasma Membrane Association by N-Acylation Governs PKG Function in *Toxoplasma gondii*" according to the mBio website). However they do not appear to have performed a dose response for auxin (or ethanol vehicle) in *Toxoplasma*. A dose response analysis would help to discern what concentrations of auxin/ethanol are toxic and could cause off target effects. Given the problems experienced in the work with the CIP2-AID knock-down, it would seem important to generate this dose response curve to provide confidence in the system.

Reviewer #2 (Remarks to the Author):

The revised manuscript by Long et al. addressed my comments in satisfactory way.

Reviewer #1 (Remarks to the Author):

“Conserved ankyrin repeat containing protein regulates conoid stability, motility, and cell invasion in *Toxoplasma*” by Dr Sibley and colleagues, *Nature Communications*.

The authors have answered most of the queries, however their response to the question about the CIP2-AID knock-down raises further questions.

The authors note that: “We tested the essentiality of CIP1 and CIP2 and found that a double deletion mutant of these two genes was viable. This surprising result led us to reassess our original finding of the auxin induced degradation of the Δ cip1 / CIP2-AID knock-down. The defect in this clone proved to be an artifact of using too high of a concentration of auxin that led to an ethanol-induced artifact.”

The authors note that “..this result only occurred in a limited number of experiments and is not a general feature of the auxin system, nor did it influence any of the other conclusions in the study.”

The authors refer back to their initial report using the auxin system in *Toxo* (which b.t.w is referenced incorrectly - the title of the 2017 paper is “Plasma Membrane Association by N-Acylation Governs PKG Function in *Toxoplasma gondii*” according to the mBio website). However they do not appear to have performed a dose response for auxin (or ethanol vehicle) in *Toxoplasma*. A dose response analysis would help to discern what concentrations of auxin/ethanol are toxic and could cause off target effects. Given the problems experienced in the work with the CIP2-AID knock-down, it would seem important to generate this dose response curve to provide confidence in the system.

Response:

We would like to thank the reviewer for pointing out the error in the reference, which we have corrected. This original reference did provide a dose response for IAA although it was in the supplement and only included a limited range of concentrations. As such, we fully agree that further testing of the parameters of this system is warranted. As such, we have included a new set of titrations that test a range of concentrations of IAA and several doses of ethanol. These tests demonstrate that up to 2,000 μ M IAA is non-toxic and that this can be combined with up to 0.2% ethanol with minimal effect. These new data are found in S3 Fig. At higher level of ethanol adverse effects are seen and this is consistent with previous reports. However, using the conditions described here the auxin system provides a rapid and non-toxic system for degradation of proteins in *Toxoplasma*.